# Team Perspective Taking and Collective Thriving in College Students’ Innovation Teams

**DOI:** 10.3390/bs14121165

**Published:** 2024-12-05

**Authors:** Hui Zhao, Mengjiao Han, Zhenzhen Wang, Bangdan Liu

**Affiliations:** Faculty of Education, Henan Normal University, Xinxiang 453007, China; 2210283148@stu.htu.edu.cn (M.H.); 2210283138@stu.htu.edu.cn (Z.W.); 2310283140@stu.htu.edu.cn (B.L.)

**Keywords:** collective thriving, team perspective taking, team trust, team reflexivity, college students

## Abstract

Team perspective taking is a process of team member empathy, motivation for other people’s ideas and feelings, and the ability to understand objectively. It can have positive impacts on teams, but the question of whether team perspective taking positively affects the sense of collective thriving exhibited by the team has not been answered, and the intrinsic mechanism underlying this influence has not been revealed. To explore the impact of team perspective taking on the collective thriving of college student innovation teams, this study constructs a chain mediation model based on theories such as the socially embedded model of thriving. A questionnaire survey was conducted to investigate 225 college student innovation teams. The results show that (1) team perspective taking, team trust, and team reflexivity are positively correlated with collective thriving. (2) Team trust and team reflexivity play separate mediating roles in the influence of team perspective taking on collective thriving. Team trust also plays a chain mediating role, and its mediating path is team perspective taking → team trust → team reflexivity → collective thriving. Team perspective taking not only has a direct effect on the collective thriving of college students’ innovation teams but also has an indirect effect through the chain mediating path of team trust and team reflexivity. This study not only further enriches the antecedent literature on collective thriving but also verifies the promoting effect of various resource factors on collective thriving at the team level and provides a theoretical reference for the activation of collective thriving.

## 1. Introduction

As the basic working unit according to the innovative talent cultivation model for college students, innovation teams have become an important force driving national innovation-driven development. The notion of a thriving innovation team has been highlighted by researchers because of the greater adaptability and better performance associated with such teams [1]. This approach builds innovative capacity through learning as well as through a focus on promoting energy, which leads to improvements in the collective ability of the team to cope with obstacles and setbacks to persist in their efforts [2]; furthermore, this approach has been shown to be correlated with team creativity [3]. The potential for collective thriving has yet to be exploited and may represent an important competitive edge with respect to team development [4]. Nonetheless, the manner in which collective thriving emerges remains largely unknown [5]. The aim of our study is to extend this research and further enrich our understanding of thriving. With China’s emphasis on college students’ innovation and entrepreneurship, colleges and universities across the country have gradually strengthened the education and training of college students’ innovation and entrepreneurship. The number of participating teams has increased explosively; at the same time, increasing attention has been given to the collective thriving of college students. What factors affect the thriving level of college students’ innovation and entrepreneurship teams? How to build a highly prosperous college student innovation and entrepreneurship team has become an urgent problem to be solved by scientific research management departments and team leaders at all levels. Therefore, investigating the factors that promote collective thriving is a timely topic that can help guide college innovation team practices.

The socially embedded model of thriving (SEMT) suggests that thriving occurs within specific work environments and involves specific resources (e.g., relational resources) [6]. At present, there are an increasing number of explorations into the mechanisms that shape perceptions of work thriving at the collective level, and fewer studies with a primary focus on research and development teams in corporate organizational contexts are being conducted, revealing the influence of work environment factors such as perceptions of organizational support, authentic leadership, and servant leadership [7,8,9] in this context. The questions of whether and how work resources impact collective thriving at the team level require further exploration. Spreitzer argued that learning and energizing occur in people’s minds and in the context of relationships as well as when performing work and observing how others perform work [10]. How to stimulate team innovation and team perspective taking has been identified as an important cognitive resource in this context [11]. Whether team perspective taking positively contributes to a team’s sense of collective thriving requires further exploration. Specifically, when team members consider others’ work motivations and ideas in an unbiased manner, the team’s information communication climate can be optimized, and work problems can be reflected upon and solved [12]. Through team perspective taking, team members can put aside their prejudices and inherent ways of thinking and consider the perspectives of others carefully, an approach that is beneficial to team dynamics and learning [11]. Team perspective taking implies positive cognition in the context of interactions, thereby promoting deep learning and positive team dynamics. Thus, this factor may contribute to collective thriving. On the basis of SEMT, this study explores the relationship between team-level perspective taking and collective thriving, which is a characteristic of highly effective teams, and investigates the theoretical mechanisms underlying this relationship.

Although SEMT explains that thriving is generated in specific work environments and resources, it is not clear which resources, especially relational resources, affect the generation of collective thriving. Whether people’s exchange of information resources plays an important role in the generation of thriving needs to be explored. Social exchange theory (SET), as a resource theory of interpersonal relations, holds that all human social activities can be reduced to an exchange relationship and that people will show their attitudes and perceptions through the interpretation of beneficial actions received by the other side. It then determines which reciprocal bank to use [13]. Thus, team perspective taking helps members think from others’ perspectives and improves their knowledge utilization, which may affect team trust; a good trusting relationship between team members will produce team belonging, which can enhance members’ sense of psychological security and facilitate the sharing of resources among members, thereby ensuring that they are energized and learn more efficiently [14]. Moreover, perspective taking involves not only facilitating information exchange but also assessing and reflecting on ideas that have been proposed more comprehensively as well as integrating different perspectives [15]. In turn, team reflexivity can enhance positive interactions within the team and establish friendly relationships among team members [6], which can enhance team members’ work dynamics and learning motivation. Simultaneously, as members grow to trust each other, they become more likely to listen to others’ opinions and engage in in-depth analyses and reflections rather than merely exerting their egos and becoming stuck in their ways [16]. Therefore, the aim of this study is to incorporate team trust and team reflexivity into the research framework for exploring the relationship between team perspective taking and collective thriving and to explore the role of the two in the chain mediation of this relationship.

Under the guidance of SEMT and SET, this study aims to establish a chain mediation model to explore the internal mechanism of team trust and team reflection in the influence of team opinion selection on collective prosperity. First, in contrast to the limited empirical research that has explored the effects of environmental factors on collective thriving, we investigate whether team perspective taking, a cognitive resource factor, is associated with collective thriving in the context of college student innovation teams with the goals of improving our understanding of the mechanisms underlying collective thriving emergence and extending SEMT at the team level. Second, by explicating how team perspective taking increases team trust and team reflexivity, which causes team collectives to become more prosperous, we elucidate the process by which team perspective taking influences teams. Finally, in the context of Chinese culture, we explore the formation mechanisms underlying collective thriving in the context of college student innovation teams and provide a theoretical framework for cross-cultural research. Our findings thus provide important insights that can promote thriving teams.

## 2. Literature Review

### 2.1. Team Perspective Taking and Collective Thriving

A team is considered to exhibit collective thriving when it collectively experiences vitality (when it exhibits energy and enthusiasm) and learning (when team members can acquire/apply new knowledge) [6]. A socially embedded, shared, group-attributable emotional and psychological state is defined as collective thriving [17], which can be intentionally utilized as team competency [1]. Previous studies have shown that servant leadership, authentic leadership, and other leadership style factors, as well as psychological experiences such as job meaning and interpersonal fairness perceptions, are significant causes of collective thriving [5,8,18]. We believe that team perspective taking can lead to the emergence of collective team thriving. Team perspective taking is a process of team member empathy, motivations for other people’s ideas and feelings, and team members’ ability to understand objectively. In the process of team perspective taking, team members objectively try to understand what other people think, motivation, and feel and why they think [15]. Spreitzer et al.’s SEMT posits that when individuals experience a specific single environment or have a specific resource, they are more likely to thrive [6]. Niessen suggested that enduring connections with others constitute an important relational resource for thriving [19]. The control of the quality of team learning associated with collective thriving requires team perspective taking to promote deep knowledge sharing and information processing. Specifically, sharing information does not imply careful processing and synthesis of ideas [15]. Team perspective taking also affects the extent to which team members can find and share information and construct information, thus reducing knowledge barriers and information bias. When perspective taking occurs, knowledge flows freely within the team, which is conducive to team learning [20]. Second, team perspective taking promotes collaborative behaviors [21], and exposing each team member to new ideas through collaboration also energizes and positively influences team members [22]. In summary, it can be inferred that team perspectives positively predict collective learning and collective vitality, i.e., collective thriving. This study thus proposes the following.

**H1.** 
*Team perspective taking is positively correlated with collective thriving.*


### 2.2. Mediating Role of Team Trust

How does team perspective taking affect the collective thriving of college student innovation teams? The information exchange relationships within the team may play an important role; according to SET, people’s attitudes and perceptions are influenced when they are the targets of beneficial behaviors from another party [13]. However, previous studies have paid less attention to the impact of these exchange relationships on collective thriving. Therefore, this study takes team relationship exchange as a starting point to explore the internal mechanism by which team perspective taking affects collective thriving. Team trust refers to an affirmative attitude held by team members toward the team [23]. Team trust includes team attributes that reflect positive expectations of other team members’ behavior and the belief that team members strive to act on their explicit and implicit commitments [24]. When members of a team consider problems from each other’s point of view and adopt their own perspective, they can promote problem solving and improve both the individual’s self-interest and the team’s common interests [25]; furthermore, the realization of these interests promotes a stronger sense of trust among team members regarding the team [26]. Team perspective taking, as a collective cognitive process, may also affect team trust. External contexts and personality traits affect individuals’ trust behavior by influencing their cognition [27]. Perspective taking, as an important concept in social cognition, may have a significant effect on interpersonal trust [28]. Perspective taking improves cognitive analyses of interpersonal interactions, thus enabling others to process cognitions more comprehensively, including not only their own interests and motivations but also those of the other person [29]; this situation allows trustors to understand the other person’s situation more comprehensively and adjust their trust in others accordingly [30]. Thus, team perspective taking may positively influence team trust.

Previous research has suggested that team trust enhances team members’ cooperation and that the resulting team collaborations lead to more extensive information sharing and expertise utilization, which can in turn facilitate knowledge acquisition and rapport building among team members [31,32]. Furthermore, team trust causes team members to be willing to invest more energy in the team and offers them sufficient psychological resources to cope with potential risks [33]. A climate of team trust also reduces employees’ perceptions toward threats, increases their sense of psychological security, eliminates the need to worry excessively about the negative consequences of violating team norms, and enhances group learning dynamics [34]. Thus, team trust is a positive predictor of collective thriving. According to the social nesting model, high-level exchange relationships, communication among team members and transpersonal thinking establish a harmonious team atmosphere and satisfy the relational needs of team members [35], thus helping establish strong trust relationships among team members and promoting team cooperation and team learning activities [36]. Thus, we propose the following hypothesis:

**H2.** 
*Team trust plays a mediating role between team perspective taking and collective thriving.*


### 2.3. Mediating Role of Team Reflexivity

In addition to team trust, team reflexivity may also play a significant role in the influence of team perspective taking on collective thriving. Team reflexivity refers to a series of logically sequenced behaviors adopted by the team to adapt to the internal and external environment, including the complete process of reflecting, proposing improvements, and making adjustments [31]. However, less research has focused on the impact of team reflexivity on collective thriving. Therefore, this study comprehensively considers the influence of team and work resources to explore the potential mediating role of team reflexivity in the relationship between team perspective taking and collective thriving. The prosocial motivation of team members is a key factor with respect to whether the team’s information processing is in-depth and systematic, and perspective taking is an embodiment of prosocial motivation members to reflect on the task at hand in a collaborative manner, thus effectively encouraging team members to reflect on the common goal [37]. This factor effectively drives team members to exhibit in-depth reflexivity and to reflect on the realization of common goals as well as existing problems [38]. Simultaneously, in teams, members’ different perspectives lead to certain differences in assessment criteria, which may cause the value of others’ viewpoints to be overlooked and lead to destructive criticism, thus impairing communication [39]. Therefore, team reflexivity requires team perspective taking. Specifically, perspective taking helps team members consider their evaluation criteria, and it facilitates the sharing and integration of different perspectives [15]. It follows that team perspective taking can promote team reflexivity among team members.

Moreover, the information exchange platform established by team reflexivity helps members understand how the team is functioning, provides them with important work-related content, such as information concerning work progress and expertise [40], and improves their understanding of the meaning of their work, thereby stimulating members’ enthusiasm for their work and ensuring that they remain energized at work. Team reflexivity is also important for team learning. First, through team reflexivity, team members can summarize and reflect on the process and results of existing team operations systematically and evaluate their own knowledge and abilities accurately [41]. Second, team reflexivity helps team members express their own insights and accept others’ perspectives, thus enabling them to fully and effectively exploit the team’s strengths, combine a variety of types of knowledge and skills within the team, promote sharing and learning among team members [42], and experience a feeling of growth and progress that can stimulate their sense of work thriving [43]. On the basis of the preceding discussion, this study suggests that team perspective taking may improve team reflexivity by affecting reciprocity and communication among group members, which in turn affects collective thriving.

**H3.** 
*Team reflexivity plays a mediating role between team perspective taking and collective thriving.*


### 2.4. Chain Mediating Roles of Team Trust and Team Reflexivity

Previous studies have focused on the effects of team perspective taking, team trust, and team reflexivity on team creativity [15,44], but there has been no in-depth study on their impact on collective thriving. On the basis of the preceding literature review, team trust and team reflexivity mediate the impacts of team perspective taking and collective thriving, respectively. The environment around an individual affects people’s behaviors and attitudes and the processing and interpretation of specific social information determine whether they should display such attitudes and behaviors [45]. Team members gain knowledge and information by trusting each other and actively participating in the decision-making process because of team trust [12], which helps the team reflect on and consider a range of important issues, such as potential partners, team scenarios, resource needs, and paths to information [23,38]. Second, team reflexivity is characterized by open discussion of the task at hand, and team trust can effectively promote social messaging with a focus on mutual recognition and respect among team members [46], thus bridging the psychological gaps among team members, bringing the team closer psychologically, dispelling team members’ worries in the context of reflexive activities, and engaging in self-expression at a deeper level [47]. It can be inferred that team trust and team reflexivity are correlated with one another and that team perspective taking may indirectly affect collective thriving through team trust and team reflexivity. According to the preceding discussion, this study proposes the following hypothesis.

**H4.** *Team trust and team reflexivity play a chain mediating role in the relationship between team perspective taking and collective thriving*.

According to the literature presented above, team perspective taking may affect team reflexivity through team trust and ultimately impact collective thriving. On the basis of SEMT, this study focused on college student innovation teams with the goals of exploring the role of team perspective taking in collective thriving and providing a reference for efforts to increase collective thriving among college student innovation teams. A diagram of the hypothesized relationship model is shown in Figure 1.

## 3. Methodology

### 3.1. Sample and Procedure

In this study, college student innovation teams that participated in the Chinese College Students’ Innovation and Entrepreneurship Project were selected as the research objects, and data were collected through an online questionnaire platform. The China College Students’ Innovation and Entrepreneurship Project is an innovative project initiated by the Ministry of Education of China, with Chinese college students as the participants, aiming to strengthen the training of college students’ innovation and entrepreneurship ability and enhance their innovation ability. In addition, this sample has good representation. These innovation teams had been working together for 1 year and focused on independently innovating and collaborating to solve problems both in their own disciplines and across disciplines. We distributed informed consent forms and questionnaires in the WeChat group of innovation training for college students and communicated with leaders to actively encourage them to participate. The participants could submit the questionnaire only after completing all the questionnaire items. The Ethics Committee of Henan Normal University supervised the study.

In this study, 932 questionnaires were distributed. After excluding invalid questionnaires that were completed in more than 600 s or less than 60 s, 851 valid questionnaires remained. The participants were 23 to 24 years old and included 558 females (65.6%) and 293 males (34.4%). The sample included 225 teams in total, and the team sizes ranged from three to five people. The sample included 12 (5.3%) six-person teams, 40 (17.8%) five-person teams, 60 (26.7%) four-person teams, and 113 (50.2%) three-person teams. The team size is 3.8 people on average. In addition, 64 (28.4%) of the teams were at least 80% male, 10 (4.4%) were between 40% and 79% male, 25 (11.1%) were between 20% and 39% male, and 126 (56.0%) were less than 20% male.

### 3.2. Measures

This research questionnaire consists of five parts: the first part is the basic information of the team, and the last four parts are the four variables measured in this research. The questionnaire was presented in Chinese, and the team perspective taking, team trust and collective thriving scales were translated from English. These scales have been applied in the Chinese context, and the Chinese translation has been verified by Chinese researchers. In this study, the Chinese translation scale was used to investigate the data and test the reliability and validity of the data, which met the requirements.

#### 3.2.1. Team Perspective Taking

Team perspective taking was measured via Grant and Berry’s revised perspective-taking scale [48], which consists of four items that ask subjects to indicate the extent to which they try to understand the perspectives of their team members, e.g., “I frequently try to adopt other team members’ perspectives”. The measure was scored on a scale ranging from 1 (“strongly disagree”) to 5 (“strongly agree”). This scale is suitable for the study of team perspective taking in college students’ teams and has good reliability and validity in previous Chinese studies [49]. Therefore, this study uses it as a scale to measure team perspective taking. In this study, the Cronbach’s α coefficient of this scale was 0.792.

#### 3.2.2. Team Trust

Team trust was measured via the Cognitive and Affective Trust Scale developed by Ng and Chua [50], which consists of eight items across two dimensions: cognitive trust (e.g., “I can trust my team members to do the majority of the team’s work”) and affective trust (e.g., “Team members tend to invest strong emotions in team task relationships”). The measure was scored on a scale ranging from 1 (“strongly disagree”) to 5 (“strongly agree”). In the past, Chinese scholars used this scale to measure team trust, and it has good reliability and validity [51]. The Cronbach’s alpha coefficient in this study was 0.863.

#### 3.2.3. Team Reflexivity

The Team Reflexivity Scale, which was adapted from Zhang and Liu [52], was used to measure team reflexivity. The scale contains 11 items across three dimensions: task reflexivity (e.g., “Before performing the task, we discuss how to complete the task”), process reflexivity (e.g., “We discuss the appropriateness of the way we use to complete the work”), and action adjustment (e.g., “We develop plans and measures to cope with changes in the environment”). The measure was scored on a scale ranging from 1 (“strongly disagree”) to 5 (“strongly agree”). Previous studies have shown that this scale is suitable for assessing the Chinese cultural background [42] and measuring the team reflexivity of college students’ innovation teams. The Cronbach’s alpha coefficient for the scale in this study was 0.874.

#### 3.2.4. Collective Thriving

The collective thriving scale developed by Porath et al. [53] was used to measure collective thriving. This scale included eight items across two dimensions, i.e., learning (e.g., “I continue to learn more and more as time goes by”) and vigor (e.g., “I feel alive and vital”); furthermore, this measure was scored on a scale ranging from 1 (“not at all”) to 5 (“completely”). In previous Chinese studies, this scale was used to measure collective thriving, and it has good reliability and validity [3]. Therefore, this study used this scale to measure the collective thriving of college students’ innovation teams, and the Cronbach’s α coefficient of this scale in this study was 0.913.

#### 3.2.5. Control Variables

In the process of data analysis, we selected team size and the proportion of males as control variables. In the study of collective scientific creativity in teams, the proportion of males cannot be ignored [54]. In addition, team size is closely related to factors such as team members’ performance and efficiency [55]. Therefore, the two variables are used as control variables to make the research results more accurate and effective.

### 3.3. Methods

In this study, statistical analysis software was used to analyze the questionnaire data. First, aggregate analysis of team data was carried out to test whether the sample data could be aggregated to the team level. The results of the validated factor analysis are shown in Table 1. Second, the four main variables are described and analyzed. The Pearson correlation coefficient was subsequently used to analyze the correlation between two variables to understand the correlation between the variables. The relevant analysis results are shown in Table 2. On the basis of correlation analysis, Model 6 of PROCESS, the SPSS V.29 plug-in provided by Hayes [56], was used to establish a chain mediation model, and the path coefficient results are shown in Figure 2. A regression analysis of Model 1~Model 3 was carried out. The results of the regression analysis are shown in Table 3. Finally, Mplus was used for the whole model path analysis, and the mediating effect was tested according to the deviation-corrected nonparametric percentile bootstrap method and 95% confidence interval, as shown in Table 4.

## 4. Results

### 4.1. Data Aggregation Efficiency Tests

Before analyzing the team-level data, it was necessary to determine whether the responses provided by the independent respondents could be aggregated to the team level. First, it was necessary to examine within-group consistency, i.e., to determine whether team members’ perceptions of the team were consistent. Second, it was necessary to examine within-group correlation, i.e., to determine whether the perceptions of different team members exhibited variability. In this study, intragroup consistency was tested by calculating the Rwg, and intragroup correlation was tested by calculating the ICC(1) and ICC(2) values. After the individual-level data were calculated, the Rwg values were 0.967 for team perspective taking, 0.973 for team trust, 0.985 for team reflexivity, and 0.982 for collective thriving, all of which were greater than 0.80, thus indicating that team members’ perceptions of the team were consistent. The ICC(1) values for team perspective taking (0.295), team trust (0.232), team reflexivity (0.215), and collective thriving (0.346) were all greater than 0.12, indicating that the between-group variance associated with all the variables was significant. The ICC(2) values for team perspective taking (0.613), team trust (0.533), team reflexivity (0.509), and collective thriving (0.667) were all greater than 0.47. This test indicated that the sample data could be aggregated at the team level.

### 4.2. Common Method Bias and Discriminant Validity Tests

As data concerning all variables were reported by team members, controls were implemented during the administration of the test by collecting data at different time points, emphasizing confidentiality and using both positive and negative scoring. To further enhance the rigor of the study, a common method bias test and a discriminant validity test were also conducted.

With respect to individual-level data, possible common method bias was examined by controlling for the unmeasured single method latent factor [57], in which context the four-variable questions loaded onto the same common method bias latent factor in addition to the variable factor to which they belonged. The models that incorporated the common method bias latent factor did not significantly outperform the four-factor model in terms of fit metrics (Δχ^2^/df = −0.031; ΔRMSEA = 0.001; ΔCFI = 0.004; ΔTLI = 0.002; ΔRMSEA = −0.006); thus, the common method bias problem was not a serious issue for this study.

Validated factor analysis was used to test the discriminant validity of the four variables included in the individual-level data. The results presented in Table 1 show that each of the fit indices of the four-factor model was good and better than those associated with the other models, thus indicating good discriminatory validity among team perspective taking, team trust, team reflexivity, and collective thriving.

### 4.3. Regression Results

The mean, standard deviation, and correlation matrix for each of the study variables are shown in Table 2. At the team level, the data showed that perspective taking, team trust, team reflexivity, and collective thriving were positively and significantly correlated with each other. Collective thriving was correlated with team size (*r* = 0.163, *p* < 0.05) and the proportion of males (*r* = 0.192, *p* < 0.01); therefore, the subsequent mediating effect test included collective thriving as a control variable in the analysis.

### 4.4. Test of the Mediating Effects

In this study, Mplus 7.4 was used for full model path analysis to test all the hypotheses [58]; Model 6 of PROCESS, the SPSS plug-in provided by Hayes [56], and team perspective taking were selected as the independent variables. With collective thriving as the dependent variable, team trust and team reflexivity as the chain mediating variables, and team size and the percentage of males as the control variables, a multisequence mediating model is established. The path coefficient results are shown in Figure 2. The regression analysis of Model 1~Model 3 was carried out. In Model 1, we conducted regressions on team trust for the control variables and team perspective taking. In Model 2, we conducted a regression on team reflexivity for the control variables, team perspective taking and team trust. In Model 3, we regress the control variables, team perspective taking, team trust, and team reflexivity on collective thriving. The whole regression equation is significant, and the regression analysis results are shown in Table 3. The model fit was χ^2^ = 802.836, df = 549, χ^2^/df = 1.462, RMSEA = 0.045, CFI = 0.944, TLI = 0.939, SRMR = 0.049, indicating a better fit.

Table 3 shows that team perspective taking can have a significant positive effect on collective thriving (B = 0.235, SE = 0.081, p < 0.01). Therefore, H1 is verified. (1) In the team perspective taking → team trust → collective thriving pathway, the positive predictive effects of team perspective taking on team trust (B = 0.608, SE = 0.057, *p* < 0.001) and team trust on collective thriving (B = 0.348, SE = 0.077, *p* < 0.001) are significant, thus suggesting that team perspective taking promotes collective thriving by enhancing students’ team trust. (2) In the team perspective taking → team reflexivity → collective thriving pathway, team perspective taking had a significant positive predictive effect on team reflexivity (B = 0.394, SE = 0.091, *p* < 0.001), and team reflexivity had a significant positive predictive effect on collective thriving (B = 0.333, SE = 0.064, *p* < 0.001), thus suggesting that collective thriving enhances collective thriving by promoting team reflexivity. (3) In the team perspective taking → team trust → team reflexivity → collective thriving pathway, the positive predictive effect of team trust on team reflexivity is significant (B = 0.424, SE = 0.082, *p* < 0.001), and the results indicate that the enhancement of team trust facilitates the formation of team reflexivity among team members. Team trust and team reflexivity have significant chain mediating effects, and team perspective taking enhances team reflexivity by increasing team trust, which in turn enhances students’ collective thriving. In conclusion, H2~H4 are supported. In addition, the direct positive predictive effect of team perspective taking on collective thriving is significant, thus indicating that team trust and team reflexivity partially mediate the relationship between team perspective taking and collective thriving (see Figure 2).

The pathway underlying the effect of team perspective taking on collective thriving is shown in Figure 1. The mediating effect was tested via bootstrap sampling, and the results revealed that the indirect effect via the path including team trust as the mediating variable was 0.211 (95% CI = [0.120, 0.337]), and the mediating effect accounted for 31.777%. The indirect effect via the path including team reflexivity as the mediating variable was 0.131 (95% CI = [0.064, 0.222]), and the mediating effect accounted for 19.729%. The indirect effect via the path including team trust and team reflexivity as the mediating variables was 0.086 (95% CI = [0.050, 0.130]), and the mediating effect accounted for 12.952%. Finally, the total of all indirect effects was 0.429 (95% CI = [0.316, 0.608]), and the total mediating effect accounted for 64.608%. These findings suggest that team trust and team reflexivity have both separate and serial mediating effects on the relationship between team perspective taking and collective thriving (Table 4).

## 5. Summary and Discussion

In this paper, the impact of team perspective taking on collective thriving and the underlying mechanism are discussed in depth in the context of China. Our study focused on college student innovation teams and analyzed the chain mediating effects of team trust and team reflexivity. This research explains the relationship between team perspective taking and collective thriving and the underlying mechanism, thereby not only providing a reference for subsequent studies of collective thriving but also offering a new method for studying the formation mechanism associated with collective thriving.

### 5.1. Theoretical Implications

This study aims to explore the relationship between team perspective taking and collective thriving in university student innovation teams, as well as the underlying mechanism involved. The findings show that there is a positive correlation between team perspective taking and collective thriving. The greater the level of team perspective taking in university student innovation teams is, the stronger the collective thriving. This study contributes to enriching SEMT to some extent and opening a new perspective for future research. Vigor and learning are closely related to the environment in which individuals live, and the greater the level of team perspective taking in innovation teams is, the greater the team’s ability to establish an environmental atmosphere that is conducive to colearning [21,49], which can promote collective thriving within the team. Every individual experiences learning and vitality; however, whether an individual can thrive depends largely on the work context in which the individual is embedded [59], including team perspective taking, which is more likely to promote thriving, as it influences the reciprocal behaviors of team members [48]. The greater the level of perspective taking is, the smaller the differences in members’ perceptions of transpersonal thinking and team reciprocity are, thus leading to relative equality in terms of the degree of transpersonal thinking exhibited by and resource inputs available to team members [60]; furthermore, reciprocity in terms of resource inputs and rewards can stimulate the motivation of team members with respect to knowledge sharing, thus triggering colearning and knowledge sharing among members [15]. When information can be fully communicated and exchanged within a team, the team can experience increased vigor and enthusiasm, thereby promoting collective team prosperity. Therefore, team perspective taking has a positive effect on the learning of team members and the team atmosphere, which can promote the collective thriving of the team. This study not only responds to the call of previous scholars to explore this topic in further detail and enrich our understanding of the antecedents of collective thriving but also verifies the contributions of various resource factors to collective thriving at the team level, thereby providing a theoretical reference for the activation of collective thriving. This study also contributes to our previous research on the factors influencing collective thriving and highlights the significant influence of work resources on collective thriving at the team level, thus leading to a new breakthrough with respect to the enhancement of collective thriving among college student innovation teams.

This study revealed that team trust mediates the relationship between team perspective taking and collective thriving and provides a new theoretical explanation mechanism for understanding the complex relationship between team perspective taking and collective thriving. This mediating effect supports SET but is also consistent with the results of previous research on the contribution of a high level of perspective taking to team trust [13,30]. That is, a higher level of team perspective taking helps team members develop trust more easily and promotes the formation of team trust, which has a positive effect on collective thriving. SET suggests that social exchange relationships represent interactions aimed at a dynamic equilibrium in which context exchange agents seek a state of equilibrium to maintain the exchange relationship [13]; accordingly, perspective taking among team members is, in essence, a social exchange relationship. Individuals experience acceptance, understanding, and reciprocal behaviors from group members when the relationship reaches equilibrium, whereas team trust is the result of establishing exchange relationships between people [61]. Second, team trust helps increase the collective thriving exhibited by college student innovation groups. On the one hand, trust among team employees can stimulate positive emotions among team members, which in turn affects individual behaviors and team climate; furthermore, a positive team atmosphere can promote knowledge and resource sharing between team members, which can increase team members’ enthusiasm for and motivation for group learning [62]. On the other hand, according to SEMT, when individuals experience an atmosphere of trust and respect, they actively engage in exploration and learning to find the actual meaning of their work [6], a situation that in turn affects job prosperity. That is, in organizations that feature high levels of team trust, individual-level perceptions of trust are more likely to translate into positive work-related outcomes [12], and employees are more likely to believe that the company is true to itself, thus helping such employees pursue their work within the organization by actively learning to improve their abilities. Therefore, teams that feature a high level of team perspective taking, communication among team members and a variety of approaches to thinking satisfy the relationship needs of team members help establish trusting relationships among team members [35]. This situation enhances team cooperation and team learning activities, thus promoting collective thriving within the team.

This study revealed that team reflexivity mediates the relationship between team perspective taking and collective thriving, which reveals the internal mechanism of team perspective taking, collective thriving and team reflexivity to a certain extent. Team perspective taking can not only directly affect the collective thriving of college students’ innovation teams but also indirectly affects collective thriving through team reflexivity. This finding is consistent with previous research [37]. When team members consider the motivations and ideas of other team members in an unbiased manner, the team communication atmosphere can be optimized, and work problems can be reflected upon and solved [12]. Perspective-taking is thus a key factor that enables team members to communicate with one another and explore team problems through shared cognition, thereby enhancing reflexivity in the context of teamwork and facilitating team reflexivity [63]. The present study revealed that team reflexivity positively predicts collective thriving, a result that is consistent with previous findings indicating that team reflexivity promotes prosperity [64]. On the one hand, team reflexivity provides team members with valuable opportunities to review and optimize the adjustments they make [6], which helps them update their knowledge resources and replenish their emotional energy in the context of teamwork, thus affecting teamwork prosperity. On the other hand, team reflexivity provides people with opportunities to share and discuss work goals, progress, and difficulties, and through such reflective activities, team members facing work difficulties can receive assistance and support, which help team members regain their motivation and confidence, replenish their positive emotional resources, and increase the vitality of learning among team members [42]. This finding shows that team perspective taking can enhance team members’ communication and exploration of team issues and influence team reflexivity, which in turn can help them replenish emotional energy within the team, increase the vigor of learning among team members, and promote collective thriving.

We also found that team trust and team reflexivity play a chain mediating role between team perspective taking and collective thriving, which contributes new knowledge to the mechanism of the impact of team perspective taking on collective thriving and further enriches the research results on the impact of team perspective taking on collective thriving. On the one hand, when a group features a good atmosphere with respect to trust, individuals derive meaning from the work environment, which influences group members’ attitudes and behaviors with respect to their work and motivates members to reflect on their team strategies and work [47]; i.e., team trust positively predicts team reflexivity. On the other hand, good team trust leads to the transfer of information regarding team belonging and identity as well as closer psychological distance among team members, thus facilitating frank communication within the team and enhancing the team’s ability to discuss their views openly in the context of program adjustments [65]. This situation makes it easier for the team to reflect more effectively on the team’s tasks and processes. Finally, team trust facilitates team members’ participation in collaboration, which can establish an atmosphere that is conducive to questioning, discussing, judging, and reflecting on perspectives within the team; reduce barriers to the open exchange of information among different team members; and enable members to access new information concerning various issues, thus making it possible for them to rethink and reflect on their own perspectives and facilitating team reflexivity [38]. Therefore, the adoption of team views is conducive to establishing trust relationships and the formation of team trust, which helps team members communicate with each other and openly discuss their views, thereby improving the level of reflexivity of the team and ultimately having a positive impact on collective prosperity.

### 5.2. Practical Implications

Team perspective taking promotes the collective thriving of college student innovation teams. Teams can organize more teamwork activities since teamwork is a type of transpersonal thinking skill that integrates different viewpoints and opinions. Through mutual collaboration and cooperation, team members can account for more viewpoints and ideas, thus enabling them to understand the problem more comprehensively, accept the ideas of other people, and improve their perspective taking [66]. Teams must also encourage their members to consider others’ feelings, and when team members communicate with each other, they should try to listen to each other’s thoughts and feelings and respect each other’s viewpoints and ideas with the goals of establishing a favorable atmosphere within the team, promoting knowledge sharing and learning, and improving collective thriving [15]. Therefore, improving the level of team perspective taking among college student teams can promote group members’ understanding of each other, reduce conflicts and contradictions, enhance the team’s learning vitality, and effectively improve the collective thriving exhibited by college student innovation teams.

Team perspective taking can influence team trust among college student innovation teams, which in turn can enhance collective thriving. First, team members can be encouraged to share achievements and honors to ensure that everyone has the opportunity to be recognized. Simultaneously, in response to challenges and mistakes, team members should also share responsibility and support and help each other. By sharing honors and responsibilities, team members can establish trust and respect for each other [67]. Second, team members should support and help each other and walk side by side in contexts featuring difficulties and challenges. By providing other team members with help and support, mutual trust can be enhanced, and closer team relationships can be established. Finally, good communication channels should be established to ensure that team members can listen to and understand each other. Team members should keep an open mind and share information, opinions, and questions with each other. Open communication helps eliminate misunderstandings and doubts and establish mutual trust [65]. Through frank communication, common goals, mutual support, and the establishment of a good team atmosphere, team members can gradually establish relationships that are characterized by mutual trust since only a trusted team can promote greater vitality and a greater level of collective thriving.

Team perspective taking can promote collective thriving by increasing team reflexivity among college student innovation teams. On the one hand, teams can set cooperative goals, thus ensuring that group members embrace the same team goals, often care about and discuss team issues, and behave toward each other in ways that can benefit both the group and the individual, thus promoting active reflexivity and open discussion among team members regarding the problems of the group [38]; such a situation is conducive to strengthening the reflective function of the team. Communication among team members is also an important way to improve learning and reflexivity. Team members can encourage, communicate with, and interact with each other; they can also organize discussions, promote activities involving sharing and cooperation within the team, and, through mutual communication, learn from each other’s experiences and ways of thinking with the goal of promoting common learning and progress. Improving a team’s reflective ability in these ways can help team members update their knowledge resources and replenish their emotional energy through teamwork, thus affecting the prosperity of the team with respect to its work.

## 6. Limitations and Directions for Future Research

This study has several limitations. First, according to SEMT, thriving is a complex psychological process that is affected by multiple factors, such as work context, relationship resources, and work resources. However, only a few influencing factors were investigated in this study, and other influencing factors, such as identification or perceived team feedback, should be considered in future studies [68]. Second, in terms of research methods, the data collected for this study were cross-sectional, and no inferences could be made regarding the causal relationships among the variables included in the study. Future research could use a longitudinal study design to improve the robustness of the research results. Third, the data concerning collective thriving exhibited by the research team were based on self-reports provided by team members and thereby referred to the self-perceptions of the members of the group; accordingly, the research results may be inaccurate to a certain extent, and the actual level of collective thriving cannot be fully and accurately displayed. Future research could consider including evaluations provided by others to ensure that the results are more objective and fairer and to enhance the reliability of the research results. Finally, this study examined the relationship between team perspective taking and collective thriving at the team level. Future research could expand beyond this context. It is necessary to develop and test cross-level models, for example, models that account for the social function perspective or employ SEMT to study the mediating roles of individual-level team inclusion, positive emotions, and other factors in the relationship between team perspective taking and collective thriving [62,69]. Such research should aim to explain the mechanism underlying the relationships between multilevel factors and collective thriving.

## 7. Conclusions

On the basis of SEMT, this study proposes a chain mediation model to explore team view relationships and collective thriving and its mechanism of action. The empirical research shows that, first, team perspective taking is significantly positively correlated with collective thriving; that is, the higher the level of perspective taking is, the easier it is for the team to achieve collective thriving. Second, team trust and team reflexivity have multiple (chain) mediating effects on the relationship between perspective taking and collective thriving via the following paths: the single mediating effect of team trust, the single mediating effect of team reflexivity, and the chain mediating effects of team trust and team reflexivity. That is, the greater the level of perspective taking exhibited by the team is, the stronger the team trust and team reflexivity are and the greater the collective thriving level is; furthermore, perspective taking promotes team reflexivity, thereby impacting the level of team trust and ultimately affecting collective thriving.

## Figures and Tables

**Figure 1 behavsci-14-01165-f001:**
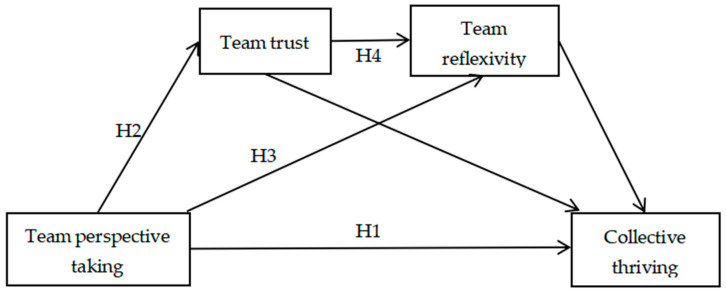
Chain mediation model of the impacts of team trust and team reflexivity on the relationship between team perspective taking and collective thriving.

**Figure 2 behavsci-14-01165-f002:**
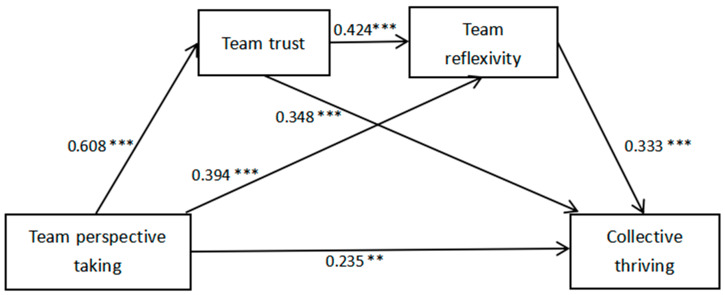
Path coefficient plot of the effect of team perspective taking on collective thriving. Note: ** *p* < 0.01, *** *p* < 0.001; the same applies below.

**Table 1 behavsci-14-01165-t001:** Results of the validated factor analysis (n = 851).

Modeling	(Math.) Factor	χ^2^	df	RMSEA	CFI	TLI	SRMR
Four-factor model	Team perspective taking; team trust; team reflexivity; collective thriving	923.201	489	0.032	0.959	0.956	0.032
Three-factor model (1)	Team perspective taking; team trust + team reflexivity; collective thriving	2119.894	492	0.062	0.848	0.837	0.057
Three-factor model (2)	Team perspective taking; team trust; team reflexivity + collective thriving	3187.690	492	0.080	0.748	0.730	0.093
Two-factor model	Team perspective taking + team trust; team reflexivity + collective thriving	3718.770	494	0.088	0.699	0.678	0.099
Single-factor model	Team perspective taking + team trust + team reflexivity + collective thriving	4906.718	495	0.102	0.588	0.560	0.105

**Table 2 behavsci-14-01165-t002:** Descriptive statistics and correlation matrix for each variable (n = 225).

	1	2	3	4	5	6
1 Team size	-					
2 Percentage of males	0.014	-				
3 Team perspective taking	0.128	0.068	-			
4 Team trust	0.123	0.091	0.557 **	-		
5 Team reflexivity	0.070	0.125	0.586 **	0.616 **	-	
6 Collective thriving	0.163 *	0.192 **	0.614 **	0.668 **	0.667 **	-
M	3.782	0.343	4.214	4.252	4.139	4.110
SD	0.922	0.437	0.298	0.319	0.255	0.326

Note: * *p* < 0.05, ** *p* < 0.01.

**Table 3 behavsci-14-01165-t003:** Results of the hypothesis testing (n = 225).

Regression Equation		Significance of Regression Coefficients				
Outcome Variable	Predictor Variable	B	S.E.	B/S.E.	BootLLCI	BootULCI
Team trust	Team size	0.060	0.054	1.102	−0.016	0.052
	Percentage of male students	0.051	0.050	1.010	−0.023	0.109
	Team perspective taking	0.608	0.057	10.673 ***	0.498	0.766
Team reflexivity	Team size	−0.030	0.049	−0.607	−0.032	0.016
	Percentage of male students	0.063	0.055	1.156	−0.012	0.085
	Team perspective taking	0.394	0.091	4.353 ***	0.184	0.469
	Team trust	0.424	0.0 82	5.192 ***	0.202	0.514
Collective thriving	Team size	0.066	0.041	1.625	−0.006	0.051
	Percentage of male students	0.115	0.038	3.041 **	0.026	0.140
	Team perspective taking	0.235	0.081	2.904 **	0.077	0.438
	Team trust	0.348	0.077	4.506 ***	0.228	0.545
	Team reflexivity	0.333	0.064	5.198 ***	0.265	0.684

Note: ** *p* < 0.01, *** *p* <0.001.

**Table 4 behavsci-14-01165-t004:** Bootstrap test of the indirect effect of team perspective taking on collective thriving.

	Indirect Effect Value	Boot SE	BootLLCI	BootULCI	Relative Mediating Effect
Total indirect effect	0.429	0.067	0.316	0.608	64.608%
Indirect effect 1	0.211	0.055	0.120	0.337	31.777%
Indirect effect 2	0.131	0.041	0.064	0.222	19.729%
Indirect effect 3	0.086	0.022	0.050	0.130	12.952%

Note: indirect effect 1 is team perspective taking → team trust → collective thriving; indirect effect 2 is team perspective taking → team reflexivity → collective thriving; and indirect effect 3 is team perspective taking → team trust → team reflexivity → collective thriving. The total indirect effect is the sum of the three indirect effects listed above.

## Data Availability

The raw data supporting the conclusions of this article will be made available by the authors upon request.

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
