# Peer review of "Team Perspective Taking and Collective Thriving in College Students’ Innovation Teams"

_behavsci, 2024, doi:10.3390/bs14121165_

Round 1
Reviewer 1 Report
Comments and Suggestions for Authors
This is really exciting work. Work like this surrounding team-based learning is necessary in today's higher education spaces and does fill a gap in the present literature.
I have a few suggestions for the manuscript. The first is that there are several theories mentioned in the introduction that are not tied in later throughout the study, likewise Social Cognitive Career Theory is mentioned late in the paper but I do not remember it in the introduction. I would suggest you limit the number of theories discussed and elaborate on those few more deeply to make a more coherent full story of your paper. Additionally, currently the introduction is largely these theories. Theories used should be presented and the introduction is a fine place to do so however you should spend a little more time giving a general overview of the study. For example, I did not know that the intent of this paper was to build a model until the results. The abstract is also quite short and you could consider adding a few more details there to help with this clarity.
Additionally, more explanation of the methods would be helpful, currently basically only the software is mentioned. You do not need to spend much space on this but it would be helpful to have sources and/or explanation to help the reader interpret figure 2. The methodology section does not have to be step by step instructions but if a reader wanted to conduct a study similar to yours, they should be able to use your methods as a starting point. Currently there is not enough description to help a reader do that.
In summary, your results, interpretation, discussion, and conclusions all seem sufficient. However, more information on the front end explaining to the reader what you were trying to do, why, and how you did it would all help this manuscript get your conclusions across more effectively.
Author Response
|
Comments 1: The first is that there are several theories mentioned in the introduction that are not tied in later throughout the study, likewise Social Cognitive Career Theory is mentioned late in the paper but I do not remember it in the introduction. I would suggest you limit the number of theories discussed and elaborate on those few more deeply to make a more coherent full story of your paper. |
|
Response 1: We sincerely appreciate your valuable comments. As you said, too many theories would make the article miscellaneous. We deleted the social cognitive career theory, Social cognitive occupational theory, social learning theory, Social information processing theory, and team-motivated information processing theory, and only kept two theories in the introduction, and the two theories were logically connected together. Please refer to the "Introduction" (pp.1-2; line 41-49) section, where we have highlighted these modifications in red [With China's emphasis on college students' innovation and entrepreneurship, colleges and universities across the country have gradually strengthened the education and training of college students' innovation and entrepreneurship. The number of participating teams has increased explosively; at the same time, increasing attention has been given to the collective thriving of college students. What factors affect the thriving level of college students' innovation and entrepreneurship teams? How to build a highly prosperous college student innovation and entrepreneurship team has become an urgent problem to be solved by scientific research management departments and team leaders at all levels.] |
|
Comments 2: Additionally, currently the introduction is largely these theories. Theories used should be presented and the introduction is a fine place to do so however you should spend slightly more time giving a general overview of the study. For example, I did not know that the intent of this paper was to build a model until the results. The abstract is also quite short and you could consider adding a few more details there to help with this clarity. |
|
Response 2: Thank Thank you for your valuable comments. According to your suggestions, we added the part of theoretical explanation and the general overview of the research in the introduction. In the last paragraph of the introduction, we made clear the purpose of this research -- to build a chain mediation model; We have also added a description of the purpose of the study to the abstract. Please refer to the "Abstract" (pp.1; line 11-13) and "Introduction" (pp.2-3; line 77-85/95-104) section, where we have highlighted these modifications in red [To explore the impact of team perspective taking on the collective thriving of college student innovation teams, this study constructs a chain mediation model based on theories such as the socially embedded model of thriving./Although SEMT explains that thriving is generated in specific work environments and resources, it is not clear which resources, especially relational resources, affect the generation of collective thriving.../Simultaneously, as members grow to trust each other, they become more likely to listen to others' opinions and engage in in-depth analyses and reflections rather than merely exerting their egos and becoming stuck in their ways.../Under the guidance of SEMT and SET, this study aims to establish a chain mediation model to explore the internal mechanism of team trust and team reflection in the influence of team opinion selection on collective prosperity.] |
|
Comments 3: Additionally, more explanation of the methods would be helpful, currently basically only the software is mentioned. You do not need to spend much space on this but it would be helpful to have sources and/or explanation to help the reader interpret figure.The methodology section does not have to be step by step instructions but if a reader wanted to conduct a study similar to yours, they should be able to use your methods as a starting point.Currently there is not enough description to help a reader do that. |
|
Response 3: Thank you for your advice. Based on your suggestions, we have added an explanation of Figure 2 and the analytical method used in the results analysis section. Please refer to the "Results" (pp.10; line 415-426) section, ,where we have highlighted these modifications in red [In this study, Mplus 7.4 was used for full model path analysis to test all the hypotheses [58], Model 6 of PROCESS, the SPSS plug-in provided by Hayes [56], and team perspective taking were selected as the independent variables. With collective thriving as the dependent variable, team trust and team reflexivity as the chain mediating variables, and team size and percentage of males as the control variables, a multisequence mediating model is established. The path coefficient results are shown in Figure 2. The regression analysis of Model 1~Model 3 was carried out. In Model 1, we conducted regressions on team trust for the control variables and team perspective taking. In Model 2, we conducted a regression on team reflexivity for the control variables, team perspective taking and team trust. In Model 3, we regress the control variables, team perspective taking, team trust, and team reflexivity on collective thriving. ] |
|
Comments 4: In summary, your results, interpretation, discussion, and conclusions all seem sufficient. However, more information on the front end explaining to the reader what you were trying to do, why, and how you did it would all help this manuscript get your conclusions across more effectively. |
|
Response 4: Thank you very much for your suggestion. We have given a fuller description of our research purpose in the abstract and introduction, so that readers can have a clearer understanding of the purpose of this research. Please refer to the "Abstract" (pp.1; line 11-21) and "Introduction" (pp.3; line 101-103) section, where we have highlighted these modifications in red [To explore the impact of team perspective taking on the collective thriving of college student innovation teams, this study constructs a chain mediation model based on theories such as the socially embedded .../The results show that (1) team perspective taking, team trust and team reflexivity are positively correlated with collective thriving. (2) Team trust and team reflexivity play separate mediating roles in the influence of team perspective taking on collective thriving.../Under the guidance of SEMT and SET, this study aims to establish a chain mediation model to explore the internal mechanism of team trust...] |
Reviewer 2 Report
Comments and Suggestions for Authors
This is an interesting and well-written paper about aspects of team working in college. The paper draws on a project in which a questionnaire was distributed to members of college student innovation teams. Hypotheses are tested based on a chain mediation model, in which the relationship between team perspective taking and collective thriving is seen as mediated by team trust and team reflexivity. The four hypotheses are supported. The findings make a range of contributions to the literature, and have potential implications for supporting the collective thriving of college students.
In many ways, this is a strong paper, and my comments are relatively light. There are, however, a few places in which more clarity or an extended explanation might be beneficial. I therefore recommend that the authors be asked to address the following points:
· The abstract could be strengthened by stating more directly (a) which area of research literature the paper contributes to and (b) the nature of the contribution that is being made.
· The Introduction could be strengthened by discussing in slightly more depth the significance of the issue of “collective thriving” in colleges. Why is this an important issue, and how has that importance been established?
· The introduction would also benefit from some clarity on the relationships between the SEMT and SET frameworks. This project clearly draws on both. Is that novel, or are the frameworks already understood to be related? It would be useful to make this clear to the reader.
· Sections 2.1-2.4 do a very good job at substantiating the hypotheses and model. But they would benefit from more critical comments about each topic in turn. What are the *shortcomings* of the literature on each of these topics, where the present paper hopes to make a contribution? Addressing this issue more directly would help set up the points made later in the discussion section.
· Section 3.1 really needs more detail about the context. What is the Chinese College Students’ Innovation and Entrepreneurship Project? What implications are there (for generalisability) for taking this as the research site? What is the target population and how representative is the sample?
· Sections 3.2.1-3.2.4 would benefit from some brief justification of *why* the four scales were chosen, in each case.
· Section 3.2 should conclude with some comments about (a) how the scales were combined into a single questionnaire instrument, and (b) in which language the instrument was distributed (English or Mandarin) and whether there were any translation issues.
· Section 4 would benefit from more signposting at the beginning. What will be covered; in which order; and how does this allow the paper to address its objectives in a systematic way?
· Table 2 contains reference to the “percentage of males” in the correlation matrix. The relevance of this should be established in the text, or else it could be simply removed. The issue seems to be inserted from nowhere; it has not previously been established as a focus for the paper.
· Section 5.1 would benefit from making the links clearer between the contributions and the shortcomings of the literature that was reviewed in sections 2.1-2.4. This need not be a complicated task; it could be done with some simple remarks in the existing text.
Author Response
|
Comments 1: The abstract could be strengthened by stating more directly (a) which area of research literature the paper contributes to and (b) the nature of the contribution that is being made. |
|
Response 1:Thank you for your valuable advice. We have added a description of the significance and value of the research at the end of the abstract, which contains the contributions in the field of literature you mentioned and the nature of the contributions made. Please refer to the "Abstract" (pp.1; line 21-24)section, where we have highlighted these modifications in red [This study not only further enriches the antecedent literature on collective thriving but also verifies the promoting effect of various resource factors on collective thriving at the team level and provides a theoretical reference for the activation of collective thriving.] |
|
Comments 2: The Introduction could be strengthened by discussing in slightly more depth the significance of the issue of “collective thriving” in colleges. Why is this an important issue, and how has that importance been established? |
|
Response 2: Thanks for your suggestion. As China attaches great importance to college students' innovation and entrepreneurship, colleges and universities across the country have gradually strengthened the education and training of college students' innovation and entrepreneurship. The number of participating teams has exploded, at the same time, more and more attention has been given to the collective thriving of college students. What factors affect the prosperity level of college students' innovation and entrepreneurship teams? How to build a highly prosperous college students' innovation and entrepreneurship team has become an urgent problem to be solved by scientific research management departments and team leaders at all levels. Therefore, it is an important and timely topic to investigate what cultivates the collective prosperity of college students' innovation team, and then guide the practice of college students' innovation team pertinently. Please refer to the " Introduction" (pp.1-2; line 41-49) section, where we have highlighted these modifications in red [With China's emphasis on college students' innovation and entrepreneurship, colleges and universities across the country have gradually strengthened the education and training of college students' innovation and entrepreneurship. The number of participating teams has increased explosively; at the same time, increasing attention has been given to the collective thriving of college students. What factors affect the thriving level of college students' innovation and entrepreneurship teams? How to build a highly prosperous college student innovation and entrepreneurship team has become an urgent problem to be solved by scientific research management departments and team leaders at all levels.] |
|
Comments 3: The introduction would also benefit from some clarity on the relationships between the SEMT and SET frameworks. This project clearly draws on both. Is that novel, or are the frameworks already understood to be related? It would be useful to make this clear to the reader. |
|
Response 3: Thank you for your very instructive suggestion. Although SEMT has explained that thriving is generated in specific work environments and resources, it is not clear which resources, especially relational resources, affect the generation of collective thriving. Whether people's exchange of information resources plays an important role in the generation of thriving needs to be explored. Social exchange theory (SET), as a resource theory of interpersonal relations, holds that all human social activities can be reduced to an exchange relationship, and people will show their attitudes and perceptions through the interpretation of beneficial actions received by the other side. It then determines which reciprocal bank to use. Please refer to the "Introduction" (pp.2; line77-84) section, where we have highlighted these modifications in red [Although SEMT explains that thriving is generated in specific work environments and resources, it is not clear which resources, especially relational resources, affect the generation of collective thriving. Whether people's exchange of information resources plays an important role in the generation of thriving needs to be explored. Social exchange theory (SET), as a resource theory of interpersonal relations, holds that all human social activities can be reduced to an exchange relationship and that people will show their attitudes and perceptions through the interpretation of beneficial actions received by the other side. It then determines which reciprocal bank to use.] |
|
Comments 4: Sections 2.1-2.4 do a very good job at substantiating the hypotheses and model. However, they would benefit from more critical comments about each topic in turn. What are the *shortcomings* of the literature on each of these topics, where the present paper hopes to make a contribution? Addressing this issue more directly would help set up the points made later in the discussion section. |
|
Response 4: Thanks to your valuable advice, we have included a fuller description of the shortcomings of previous studies and the purpose of this study in sections 2.1-2.4. Please refer to the "Literature review" (pp.3-5; line121-125/146-153/187-195/227-229) section, where we have highlighted these modifications in red [Previous studies have shown that servant leadership, authentic leadership, and other leadership style factors, as well as psychological experiences such as job meaning and interpersonal fairness perceptions, are important causes of collective thriving.../How does team perspective taking affect the collective thriving of college student innovation teams? The information exchange relationships within the team may play an important role.../In addition to team trust, team reflexivity may also play a significant role in the influence of team perspective taking on collective thriving.../Previous studies have focused on the effects of team perspective taking, team trust, and team reflexivity on team creativity, but there has been no in-depth study on their impact on collective thriving...] |
|
Comments 5: Section 3.1 truly needs more detail about the context. What is the Chinese College Students’ Innovation and Entrepreneurship Project? What implications are there (for generalisability) for taking this as the research site? What is the target population and how representative is the sample? |
|
Response 5: Thank you for your suggestion. The Chinese College Students Innovation and Entrepreneurship Project is an innovative project initiated by the Ministry of Education of China, aiming to strengthen the training of college students' innovation and entrepreneurship ability and enhance their innovation ability. Since the research object of this study is the college student innovation team, the participating teams of this project are in line with the theme of this study and are representative, so the teams participating in the innovation and entrepreneurship project of Chinese college students are taken as the samples of this study. The target population is college students in China, and the sample of the study has a good representative. We have given a more detailed description of Chinese college student innovation and entrepreneurship projects in Section 3.1. Please refer to the "Sample and procedure" (pp.6; line 278-283) section, where we have highlighted these modifications in red [The China College Students' Innovation and Entrepreneurship Project is an innovative project initiated by the Ministry of Education of China, with Chinese college students as the participants, aiming to strengthen the training of college students' innovation and entrepreneurship ability and enhance their innovation ability. In addition, this sample has good representation.] |
|
Comments 6: Sections 3.2.1-3.2.4 would benefit from some brief justification of *why* the four scales were chosen, in each case. |
|
Response 6: Thanks for your suggestions, we have a brief explanation of the reasons for choosing to use these four scales in sections 3.2.1-3.2.4. Please refer to the "Measures" (pp.7-8; line 313-316/324-325/334-336/343-345) section, where we have highlighted these modifications in red [This scale is suitable for the study of team perspective taking in college students' teams and has good reliability and validity in previous Chinese studies.../In the past, Chinese scholars used this scale to measure team trust, and it has good reliability and validity./Previous studies have shown that this scale is suitable for assessing the Chinese cultural background and for measuring the team reflexivity of college students' innovation teams./In previous Chinese studies, this scale was used to measure collective thriving, and it has good reliability and validity.] |
|
Comments 7: Section 3.2 should conclude with some comments about (a) how the scales were combined into a single questionnaire instrument, and (b) in which language the instrument was distributed (English or Mandarin) and whether there were any translation issues. |
|
Response 7: Thanks for your suggestion, this research questionnaire consists of five parts, the first part is the basic information of the team, and the last four parts are the four variables measured in this research. The questionnaire was presented in Chinese, in which the team perspective taking, team trust and collective thriving scales were translated from English. These scales have been applied in the Chinese context, and the Chinese translation has been verified by Chinese researchers. In this study, the Chinese translation scale was used to investigate the data and test the reliability and validity of the data, which met the requirements. In the main text, we have added reference identification which we explain in the research Measures section Please refer to the "Measures" (pp.6-7; line 301-307) section, where we have highlighted these modifications in red [This research questionnaire consists of five parts: the first part is the basic information of the team, and the last four parts are the four variables measured in this research. The questionnaire was presented in Chinese, and the team perspective-taking, team trust and collective thriving scales were translated from English. These scales have been applied in the Chinese context, and the Chinese translation has been verified by Chinese researchers. In this study, the Chinese translation scale was used to investigate the data and test the reliability and validity of the data, which met the requirements.../] |
|
Comments 8: Section 4 would benefit from more signposting at the beginning. What will be covered; in which order; and how does this allow the paper to address its objectives in a systematic way? |
|
Response 8: Thanks to your valuable suggestions, we have added more descriptions of the steps and contents of the analysis method before the Section 4, the Section Results to make the analysis method more clear and logical. Please refer to the "Methods" (pp.8; line 354-366) section, where we have highlighted these modifications in red [In this study, statistical analysis software was used to analyze the questionnaire data. First, aggregate analysis of team data was carried out to test whether the sample data could be aggregated to the team level. The results of the validated factor analysis are shown in Table 1. Second, the four main variables are described and analyzed. The Pearson correlation coefficient was subsequently used to analyze the correlation between two variables to understand the correlation between the variables. The relevant analysis results are shown in Table 2. On the basis of correlation analysis, Model 6 of PROCESS, the SPSS plug-in provided by Hayes, is used to establish a chain mediation model, and the path coefficient results are shown in Figure 2. A regression analysis of Model 1~Model 3 was carried out. The results of the regression analysis are shown in Table 3. Finally, Mplus was used for the whole model path analysis, and the mediating effect was tested according to the deviation-corrected nonparametric percentile bootstrap method and 95% confidence interval, as shown in Table 4.] |
|
Comments 9: Table 2 contains reference to the “percentage of males” in the correlation matrix. The relevance of this should be established in the text, or else it could be simply removed. The issue seems to be inserted from nowhere; it has not previously been established as a focus for the paper. |
|
Response 9: Thanks to your suggestion, we added the percentage of males correlation in the section 4.3 Regression result. A description of the reasons for controlling the percentage of males was added in 3.2.5 Control variables section. Please refer to the "Control variables" (pp.7; line 347-352) and "Regression results" (pp.9; line 408-410) section, where we have highlighted these modifications in red [In the process of data analysis, we selected team size and the proportion of males as control variables. In the study of collective scientific creativity in teams, the proportion of males cannot be ignored [54]. In addition, team size is closely related to factors such as team members' performance and efficiency [55]. Therefore, the two variables are used as control variables to make the research results more accurate and effective./Collective thriving was correlated with team size (r= 0.163, p < 0.05) and the proportion of males (r = 0.192, p < 0.01); therefore, the subsequent mediating effect test included collective thriving as a control variable in the analysis.] |
|
Comments 10: Section 5.1 would benefit from making the links clearer between the contributions and the shortcomings of the literature that was reviewed in sections 2.1-2.4. This need not be a complicated task; it could be done with some simple remarks in the existing text. |
|
Response 10: Thanks for your valuable suggestion, we have improved and supplemented the theoretical value based on the contributions and shortcomings described in sections 2.1-2.4 Please refer to the "Theoretical implications" (pp.12-14; line 490-496/523-525/553-557/580-583) section, where we have highlighted these modifications in red [This study aims to explore the relationship between team perspective taking and collective thriving in university student innovation teams, as well as the underlying mechanism involved.../and provides a new theoretical explanation mechanism for understanding the complex relationship between team perspective taking and collective thriving..../This study revealed that team reflexivity mediates the relationship between team perspective taking and collective thriving.../which contributes new knowledge to the mechanism of the impact of team perspective taking on collective thriving and further enriches the research results on the impact of team perspective taking on collective thriving.] |
Reviewer 3 Report
Comments and Suggestions for Authors
1. I think the term “team perspective taking” needs a short definition at the beginning of the abstract.
2. A brief summary of the results needs to be given in the abstract.
3. The hypothesis numbers should be added in Fig. 1.
4. In the methodology section, it would be helpful to give an impression on what kinds of exemplary problems the teams had been working.
Author Response
|
Comments 1: I think the term “team perspective taking” needs a short definition at the beginning of the abstract. |
|
Response 1:Thanks to your suggestion, we have included the definition of team perspective taking in the abstract. Please refer to the "Abstract" (pp.1; line 7-8) section, where we have highlighted these modifications in red [Team perspective taking is a process of team member empathy, motivation for other people's ideas and feelings, and ability to understand objectively.] |
|
Comments 2: A brief summary of the results needs to be given in the abstract. |
|
Response 2: Thanks for your suggestion, we have included a summary of the findings in the abstract. Please refer to the "Abstract" (pp.1; line 14-21) section, where we have highlighted these modifications in red [The results show that (1) team perspective taking, team trust and team reflexivity are positively correlated with collective thriving. (2) Team trust and team reflexivity play separate mediating roles in the influence of team perspective taking on collective thriving. Team trust also plays a chain mediating role, and its mediating path is team perspective taking → team trust → team reflexivity → collective thriving. Team perspective taking not only has a direct effect on the collective thriving of college students' innovation teams but also has an indirect effect through the chain mediating path of team trust and team reflexivity.] |
|
Comments 3: The hypothesis numbers should be added in Fig. 1. |
|
Response 3: Thanks to your suggestion, we have included hypothesis numbers in Fig. 1 Please refer to the "Fig. 1" (pp.6; line 260-269), where we have highlighted these modifications in red. |
|
Comments 4: In the methodology section, it would be helpful to give an impression on what kinds of exemplary problems the teams had been working. |
|
Response 4: Thanks to your valuable suggestions, we have enriched the description of more steps and contents of the analysis method before section 4 to make the analysis method more clear and logical. Please refer to the "Methods" (pp.8; line 354-366) section, where we have highlighted these modifications in red [In this study, statistical analysis software was used to analyze the questionnaire data. First, aggregate analysis of team data was carried out to test whether the sample data could be aggregated to the team level. The results of the validated factor analysis are shown in Table 1. Second, the four main variables are described and analyzed. The Pearson correlation coefficient was subsequently used to analyze the correlation between two variables to understand the correlation between the variables. The relevant analysis results are shown in Table 2. On the basis of correlation analysis, Model 6 of PROCESS, the SPSS plug-in provided by Hayes [56], is used to establish a chain mediation model, and the path coefficient results are shown in Figure 2. A regression analysis of Model 1~Model 3 was carried out. The results of the regression analysis are shown in Table 3. Finally, Mplus was used for the whole model path analysis, and the mediating effect was tested according to the deviation-corrected nonparametric percentile bootstrap method and 95% confidence interval, as shown in Table 4.] |